# GRAPH NEURAL NETWORKS FOR LEARNING EQUIVARIANT REPRESENTATIONS OF NEURAL NETWORKS

**Miltiadis Kofinas**[1][*]   **Boris Knyazev**[2]   **Yan Zhang**[2]   **Yunlu Chen**[3]

**Gertjan J. Burghouts**[4]   **Efstratios Gavves**[1]   **Cees G. M. Snoek**[1]   **David W. Zhang**[1][*]

[1]University of Amsterdam   [2]Samsung - SAIT AI Lab, Montreal
[3]Carnegie Mellon University   [4]TNO

## ABSTRACT

Neural networks that process the parameters of other neural networks find applications in domains as diverse as classifying implicit neural representations, generating neural network weights, and predicting generalization errors. However, existing approaches either overlook the inherent permutation symmetry in the neural network or rely on intricate weight-sharing patterns to achieve equivariance, while ignoring the impact of the network architecture itself. In this work, we propose to represent neural networks as computational graphs of parameters, which allows us to harness powerful graph neural networks and transformers that preserve permutation symmetry. Consequently, our approach enables a single model to learn from neural graphs with diverse architectures. We showcase the effectiveness of our method on a wide range of tasks, including classification and editing of implicit neural representations, predicting generalization performance, and learning to optimize, while consistently outperforming state-of-the-art methods. The source code is open-sourced at `https://github.com/mkofinas/neural-graphs`.

## 1   INTRODUCTION

How can we design neural networks that themselves take *neural network parameters* as input? This would allow us to make inferences *about* neural networks, such as predicting their generalization error (Unterthiner et al., 2020), generating neural network weights (Schürholt et al., 2022a), and classifying or generating implicit neural representations (Dupont et al., 2022) without having to evaluate them on many different inputs. For simplicity, let us consider a deep neural network with multiple hidden layers. As a naïve approach, we can simply concatenate all flattened weights and biases into one large feature vector, from which we can then make predictions as usual. However, this overlooks an important structure in the parameters: neurons in a layer can be *reordered* while maintaining exactly the same function (Hecht-Nielsen, 1990). Reordering neurons of a neural network means permuting the preceding and following weight matrices accordingly. Ignoring the permutation symmetry will typically cause this model to make different predictions for different orderings of the neurons in the input neural network, even though they represent exactly the same function.

In general, accounting for symmetries in the input data improves the learning efficiency and underpins the field of geometric deep learning (Bronstein et al., 2021). Recent studies (Navon et al., 2023; Zhou et al., 2023a) confirm the effectiveness of equivariant layers for *parameter spaces* (the space of neural network parameters) with specially designed weight-sharing patterns. These weight-sharing patterns, however, require manual adaptation to each new architectural design. Importantly, a single model can only process neural network parameters for a single fixed architecture. Motivated by these observations, we take an alternative approach to address the permutation symmetry in neural networks: we present the *neural graph* representation that connects neural network parameters similar to the computation graph (see Figure 1). By explicitly integrating the graph structure in our neural network, a single model can process *heterogeneous* architectures, *i.e.* architectures with varying computational graphs, including architectures with different number of layers, number of hidden dimensions, non-linearities, and different network connectivities such as residual connections.

---

[*]Joint first and last authors. Correspondence to: `mkofinas@gmail.com`, `david0w0zhang@gmail.com`

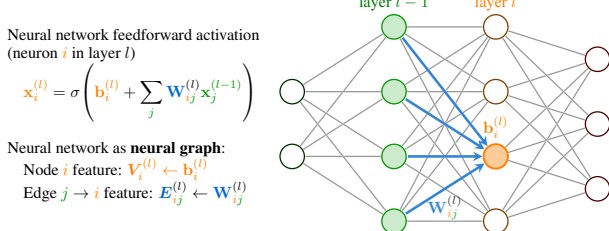

Neural network feedforward activation
(neuron $i$ in layer $l$)

$$\mathbf{x}_i^{(l)} = \sigma\left(\mathbf{b}_i^{(l)} + \sum_j \mathbf{W}_{ij}^{(l)} \mathbf{x}_j^{(l-1)}\right)$$

Neural network as **neural graph**:
Node $i$ feature: $\boldsymbol{V}_i^{(l)} \leftarrow \mathbf{b}_i^{(l)}$
Edge $j \rightarrow i$ feature: $\boldsymbol{E}_{ij}^{(l)} \leftarrow \mathbf{W}_{ij}^{(l)}$

Figure 1: Representing a neural network and its weights as a *neural graph*. We assign neural network parameters to graph features by treating biases $\mathbf{b}_i$ as corresponding node features $\boldsymbol{V}_i$, and weights $\mathbf{W}_{ij}$ as edge features $\boldsymbol{E}_{ij}$ connecting the nodes in adjacent layers.

We make the following contributions. First, we propose a simple and efficient representation of neural networks as *neural graphs*, which ensures invariance to neuron symmetries. The perspective of permuting neurons rather than parameters makes our model conceptually much simpler than prior work. We detail various choices on how the neural graph can encode a neural network, including the novel concept of "probe features" that represent the neuron activations of a forward pass. Second, we adapt existing graph neural networks and transformers to take neural graphs as input, and incorporate inductive biases from neural graphs. In the context of geometric deep learning, neural graphs constitute a new benchmark for graph neural networks. Finally, we empirically validate our proposed method on a wide range of tasks, and outperform state-of-the-art approaches by a large margin.

## 2 NEURAL NETWORKS AS NEURAL GRAPHS

Two aspects determine a neural network's behavior: the parameters, and the architecture that defines how the parameters are used. Prior work has only modeled one of these, namely the parameters, while assuming a fixed architecture. This limits the existing models to a single input architecture; any changes (regardless of whether they affect the parameters or not, such as through skip-connections) cannot be accounted for. These changes can also alter the network's symmetry group, which requires a complete redesign for every new architecture. Our approach therefore considers both the parameters as well as the architecture that specifies the computational instructions of the neural network.

Our method is straightforward: we represent the input neural network as a graph, with its nodes corresponding to individual neurons in the neural network and its edges corresponding to connections between neurons. The weights of the input neural network determine the edge features and the biases determine the node features. We refer to this as the *neural graph*. Importantly, the natural symmetries in these graphs correspond exactly to the neuron permutation symmetries in neural networks: when we permute the nodes of the neural graph, the adjacency matrix is permuted in a way such that the connections between the same neurons remain the same. Therefore, different neuron permutations of the input neural network – which correspond to the same function – result in the same graph as well, which is precisely what we desire[1]. Another benefit of the neural graph representation is that there is an extensive body of prior research on graph neural networks that we can make use of to obtain powerful models. Graph neural networks and transformers naturally exhibit equivariance to the permutation symmetries of graphs, and therefore our neural graphs that represent neural networks.

In the following section, we explain how we convert Multi-Layer Perceptrons (MLPs) into neural graphs. Afterwards, we show how to extend the conversion to Convolutional Neural Networks (CNNs). Finally, we show how our method can process heterogeneous architectures.

### 2.1 MLPs AS GRAPHS

We first outline the procedure for constructing a neural graph $\mathcal{G} = (\boldsymbol{V}, \boldsymbol{E})$ with node features $\boldsymbol{V} \in \mathbb{R}^{n \times d_{\boldsymbol{V}}}$ and edge features $\boldsymbol{E} \in \mathbb{R}^{n \times n \times d_{\boldsymbol{E}}}$. Here, $n$ denotes the total number of nodes in the graph, and $d_{\boldsymbol{V}}, d_{\boldsymbol{E}}$ denote the number of node and edge features, respectively. Consider an MLP with $L$ fully connected layers. The weight matrices for this MLP are $\{\mathbf{W}^{(1)}, \ldots, \mathbf{W}^{(L)}\}$, and the biases are $\{\mathbf{b}^{(1)}, \ldots, \mathbf{b}^{(L)}\}$. Each weight matrix $\mathbf{W}^{(l)}$ has dimensions $d_l \times d_{l-1}$ and each bias $\mathbf{b}^{(l)}$ has dimensions $d_l$. The total number of nodes is then given by $n = \sum_{l=0}^{L} d_l$, where $d_0$ is the dimension of the input. We define the edge and node feature matrices containing the weights and biases as

---

[1]Neurons in the input and output layers are typically an exception to this, as they are not freely exchangeable without changing the underlying function. Section 3 addresses this through positional embeddings.

follows:

$$
\boldsymbol{E} = \begin{pmatrix}
\begin{matrix}
\mathbf{W}^{(1)\top} & & & \ddots & & \\
 & \mathbf{W}^{(2)\top} & & & \mathbf{0} & \\
 & & \ddots & & & \ddots \\
 & \mathbf{0} & & \ddots & & \\
 & & & & \mathbf{W}^{(L)\top} & \\
 & & \ddots & & &
\end{matrix}
\end{pmatrix}
\begin{matrix} d_0 \\ d_1 \\ d_2 \\ \vdots \\ d_L \end{matrix}
\quad , \quad
\boldsymbol{V} = \begin{pmatrix}
\mathbf{0}_{d_0} \\ \mathbf{b}^{(1)} \\ \mathbf{b}^{(2)} \\ \vdots \\ \mathbf{b}^{(L)}
\end{pmatrix}
\tag{1}
$$

where $\mathbf{W}^{(l)\top}$ denotes the transposed weight matrix. The edge features form a sparse block matrix where the sparsity pattern depends on the number of nodes per layer: the first diagonal block has size $d_0$, the second $d_1$, and so on. We can verify that putting $\mathbf{W}^{(l)}$ as the first off-diagonal blocks means that they have the expected size of $d_l \times d_{l-1}$. The first $d_0$ node features in $\boldsymbol{V}$ are set to 0 to reflect the fact that there are no biases at the input nodes. We can show that our neural graph has a one-to-one correspondence to the neural network's computation graph. This ensures that distinct MLPs get mapped to distinct neural graphs.

Similar to the example above, many applications of neural networks in parameter space only include the neural network parameters as input. For such applications, an MLP has scalar weights $\mathbf{W}_{ij}^{(l)} \in \mathbb{R}$ and biases $\mathbf{b}_i^{(l)} \in \mathbb{R}$ comprising the elements of $\boldsymbol{E}, \boldsymbol{V}$, resulting in one-dimensional features, *i.e.* $d_{\boldsymbol{V}} = d_{\boldsymbol{E}} = 1$. Depending on the task at hand, however, we have the flexibility to incorporate additional edge and node features. We explore some examples of this in Section 2.4.

## 2.2 CNNs AS GRAPHS

So far, we have only described how to encode basic MLPs as graphs. We now address how to generalize the graph representation to alternative network architectures, namely convolutional networks. To make the exposition of our method clear, we will use the following interpretation of convolutional layers and CNNs. Convolutional layers take as input a multi-channel input image (*e.g.* an RGB image has $C = 3$ channels) or a multi-channel feature map, and process it with a filter bank, *i.e.* a collection of convolutional kernels – often termed filters. Each filter results in a single-channel feature map; applying all filters in the filter bank results in a collection of feature maps, which we concatenate together in a multi-channel feature map. CNNs, in their simplest form, are a stack of convolutional layers mixed with non-linearities in between.

Permutation symmetries in a CNN work similarly to an MLP; permuting the filters in a layer while simultaneously permuting the channels of each filter in the subsequent layer effectively cancels out the permutations, shown visually in Figure 5 in Appendix C.2. Under the aforementioned interpretation, single-channel slices of a multi-channel feature map (or single-channel slices of the input) correspond to nodes in our neural graph. Each node is connected via edges incoming from a particular convolutional kernel from the filter bank. By treating each channel as a node, our CNN neural graph respects the permutation symmetries of the CNN.

We now describe the neural graph representation for each component in a CNN (with more details in Appendix C.2). As a working example, let us consider a CNN with $L$ convolutional layers. It consists of filters $\{\mathbf{W}^{(l)}\}$ and biases $\{\mathbf{b}^{(l)}\}$ for layers $l \in \{1, \ldots, L\}$, where $\mathbf{W}^{(l)} \in \mathbb{R}^{d_l \times d_{l-1} \times w_l \times h_l}, \mathbf{b}^{(l)} \in \mathbb{R}^{d_l}$, and $w_l, h_l$ denote the width and the height of kernels at layer $l$.

**Convolutional layers.** Convolutional layers are the core operation in a convolutional network. Since channels in a CNN correspond to nodes in the neural graph, we can treat the biases the same way as in an MLP, namely as node features – see Equation (1). The kernels, however, cannot be treated identically due to their spatial dimensions, which do not exist for MLP weights. To resolve this, we represent the kernels by flattening their spatial dimensions to a vector. To ensure spatial self-consistency across kernels of different sizes, we first zero-pad all kernels to a maximum size $s = (w_{\max}, h_{\max})$, and then flatten them. This operation allows for a unified representation across different kernel sizes; we can process all kernels with the same network. The maximum kernel size is

chosen as a hyperparameter per experiment; this operation is visualized in Figure 6 in Appendix C.2. After this operation, the kernels can be treated as a multi-dimensional equivalent of linear layer weights. We construct the edge features matrix similarly to Equation (1); the only difference is that this matrix no longer has scalar features. Instead, we have $\boldsymbol{E} \in \mathbb{R}^{n \times n \times d_{\boldsymbol{E}}}$, with $d_{\boldsymbol{E}} = w_{\max} \cdot h_{\max}$.

**Flattening layer.** CNNs are often tasked with predicting a single feature vector per image. In such cases, the feature maps have to be converted to a single feature vector. Modern CNN architectures (He et al., 2016) perform adaptive pooling after the last convolutional layer, which pools the whole feature map in a single feature vector, while traditional CNNs (Simonyan & Zisserman, 2015) achieved that by flattening the spatial dimensions of the feature maps. The downside of the latter approach is that CNNs are bound to a specific input resolution and cannot process arbitrary images. Our neural graph is not bound to any spatial resolution, and as such, its construction does not require any modifications to integrate adaptive pooling. While the CNNs in all our experiments employ adaptive pooling, we also propose two mechanisms to address traditional flattening, which we discuss in Appendix C.2.

**Linear layers.** Linear layers are often applied after flattening or adaptive pooling to produce the final feature vector representation for an image. The most straightforward way to treat linear layers in a CNN is in the exact same fashion as in an MLP. The downside of this approach is that linear layers and convolutional layers have separate representations, as their $d_{\boldsymbol{E}}$ will typically differ. An alternative is to treat linear layers as $1 \times 1$ convolutions, which allows for a unified representation between linear and convolutional layers. When treated as convolutions, the linear layers are padded to the maximum kernel size and flattened. In our experiments, we explore both options, and choose the most suitable via hyperparameter search.

## 2.3 MODELLING HETEROGENEOUS ARCHITECTURES

One of the primary benefits of the neural graph representation is that it becomes straightforward to represent varying network architectures that can all be processed by the same graph neural network. Notably, we do not require any changes to accommodate a varying number of layers, number of dimensions per layer, or even completely different architectures and connectivities between layers. When dealing with a single architecture, we can opt to ignore certain architectural components that are shared across instances, as our method – and related methods – can learn to account for them during training. These include activation functions and residual connections. Thus, we will now describe how we can incorporate varying non-linearities and residual connections in heterogeneous architectures of CNNs or MLPs.

**Non-linearities.** Non-linearities are functions applied elementwise to each neuron, and can thus be encoded as node features. We create embeddings for a list of common activation functions and add them to the node features.

**Residual connections.** Residual connections are an integral component of modern CNN architectures. A residual connection directly connects the input of a layer to its output as $\mathbf{y} = f(\mathbf{x}) + \mathbf{x}$. Incorporating residual connections in our neural graph architecture is straightforward, since we can include edges from each sender node in $\mathbf{x}$ to the respective receiving node in $\mathbf{y}$. Since residual connections can be rewritten as $\mathbf{y} = f(\mathbf{x}) + \mathbf{I}\mathbf{x}$, where $\mathbf{I}$ is the identity matrix, the edge features have a value of 1 for each neuron connected.

**Transformers.** The feedforward component of transformers (Vaswani et al., 2017) can be treated as an MLP and its neural graph follows a similar schema as previously outlined. We explain the conversion rules for normalization layers in Appendix C.3 and the conversion rules for multi-head self-attention in Appendix C.4. After individually converting the parts, we can compose them into a neural graph for transformers.

## 2.4 NODE AND EDGE REPRESENTATION

Our neural graph representation gives us the flexibility to choose what kinds of data serve as node and edge features. Though we mainly focus on weights and biases, there are also other options that we can use, for example we use the gradients in a learning to optimize setting.

**Edge direction.** Our basic encoding only considers the forward pass computations of the neural network, yielding a directed acyclic graph. To facilitate information flow from later layers back

to earlier layers we can add reversed edges to the neural graph. Specifically, we include $\boldsymbol{E}^\top$ as additional edge features. Similarly, we can also include $\boldsymbol{E} + \boldsymbol{E}^\top$ as extra features representing undirected features.

**Probe features.** Humans tend to interpret complicated functions by probing the function with a few input samples and inspecting the resulting output. We give the graph neural network a similar ability by adding extra features to every node that correspond to specific inputs. In particular, we learn a set of sample input values that we pass through the input neural network and retain the values for all the intermediate activations and the output. For example, consider the simple input neural network $f(\boldsymbol{x}) = \mathbf{W}^{(2)}\alpha\big(\mathbf{W}^{(1)}\boldsymbol{x} + \mathbf{b}^{(1)}\big) + \mathbf{b}^{(2)}$ for which we acquire an extra node feature:

$$\boldsymbol{V}_{\text{probe}} = \Big(\boldsymbol{x}, \alpha\Big(\mathbf{W}^{(1)}\boldsymbol{x} + \mathbf{b}^{(1)}\Big), f(\boldsymbol{x})\Big)^\top, \tag{2}$$

per $\boldsymbol{x} \in \{\boldsymbol{x}_m\}_{m=1,\dots,M}$ where $\{\boldsymbol{x}_m\}_{m=1,\dots,M}$ is a set of learned input values. For simplicity, we use the same set for all neural graphs. The features in Equation 2 are then included as additional node features. Notably, probe features are invariant to *all* augmentations on the input neural network's parameters as long as it maintains the exact same function for the output and hidden layers.

**Normalization.** Existing works on parameter space networks (Navon et al., 2023; Zhou et al., 2023a) perform feature normalization by computing the mean and standard deviation separately for each neuron in the training set. This operation, however, violates the neuron symmetries; since neurons can be permuted, it becomes essentially arbitrary to normalize neurons across neural networks. We propose a simple alternative that respects permutation equivariance: we compute a single mean and standard deviation for each layer (separately for weights and biases), and use them to standardize our neural graph, *i.e.* $\hat{\mathbf{W}}^{(l)} = \Big(\mathbf{W}^{(l)} - \mu_W^{(l)}\Big)/\sigma_W^{(l)}, \hat{\mathbf{b}}^{(l)} = \Big(\mathbf{b}^{(l)} - \mu_b^{(l)}\Big)/\sigma_b^{(l)}, l \in \{1, \dots, L\}$ .

**Positional embeddings.** Before processing the neural graph, we augment each node with learned positional embeddings. To maintain the permutation symmetry in the hidden layers, nodes corresponding to the same intermediate layer share the same positional embedding. Although this layer information is implicitly available in the adjacency matrix, using positional embeddings allows immediate identification, eliminating the need for multiple local message-passing steps. However, we distinguish between the symmetries of input and output nodes and those of hidden nodes. In a neural network, rearranging the input or output nodes generally alters the network's underlying function. In contrast, the graph representation is indifferent to the order of input and output nodes. To address this discrepancy, we introduce unique positional embeddings for each input and output node, thereby breaking the symmetry between and enabling GNNs and transformers to differentiate between them.

## 3 Learning with Neural Graphs

Graph neural networks (GNNs) and transformers are equivariant with respect to the permutation symmetries of graphs. We present one variant of each and adapt them for processing neural graphs.

**GNN.** Graph neural networks (Scarselli et al., 2008; Kipf & Welling, 2017) in the form of message passing neural networks (Gilmer et al., 2017) apply the same local message passing function at every node. While various GNN variants exist, only few of them are designed to accommodate edge features, and even fewer update these edge features in hidden layers. Updating edge features is important in our setting since our primary features reside on the edges; when per-weight outputs are required, they become imperative. We choose PNA (Corso et al., 2020) as our backbone, a state-of-the-art graph network that incorporates edge features. However, PNA does not update its edge features. To address this gap, we apply a common extension to it by updating the edge features at each layer using a lightweight neural network $\phi_e$:

$$\mathbf{e}_{ij}^{(k+1)} = \phi_e^{(k+1)}\Big(\Big[\mathbf{v}_i^{(k)}, \mathbf{e}_{ij}^{(k)}, \mathbf{v}_j^{(k)}\Big]\Big), \tag{3}$$

where $k$ is the layer index in our network. This ensures that the edge features are updated per layer based on incident node features and produce a representation that depends on the graph structure.

Navon et al. (2023) suggest that the ability to approximate the forward pass of the input neural network can be indicative of expressive power. Part of the forward pass consists of the same operation for each edge: multiply the weight with the incoming activation. Motivated by the insights on algorithmic alignment by Xu et al. (2020), we adapt the message-passing step to include this multiplicative

interaction between the node and edge features. In particular, we apply FiLM to the message passing step (Perez et al., 2018; Brockschmidt, 2020):

$$\mathbf{m}_{ij} = \phi_{\texttt{scale}}(\mathbf{e}_{ij}) \odot \phi_m([\mathbf{v}_i, \mathbf{v}_j]) + \phi_{\texttt{shift}}(\mathbf{e}_{ij}). \tag{4}$$

Note that this differs from the FiLM-GNN (Brockschmidt, 2020) in that we compute the scaling factors based on the edge features and not based on the adjacent node's features.

**Transformer.** The transformer encoder (Vaswani et al., 2017) can be seen as a graph neural network that operates on the fully connected graph. Similar to GNNs, the original transformer encoder and common variants do not accommodate edge features. We use the transformer variant with relational attention (Diao & Loynd, 2023) that adds edge features to the self-attention computation. The relational transformer is already equipped with updates to the edge features. As per GNNs, we further augment the transformer with modulation to enable multiplicative interactions between the node and edge features. In particular, we change the update to the value matrix in the self-attention module:

$$\mathbf{v}_{ij} = \left(\mathbf{W}_{\texttt{scale}}^{\text{value}}\mathbf{e}_{ij}\right) \odot \left(\mathbf{W}_n^{\text{value}}\mathbf{v}_j\right) + \mathbf{W}_{\texttt{shift}}^{\text{value}}\mathbf{e}_{ij}. \tag{5}$$

## 4 EXPERIMENTS

We assess the effectiveness of our approach on a diverse set of tasks requiring neural network processing, with either per-parameter outputs (equivariant) or a global output (invariant). We refer to the graph network variant of our method as NG-GNN (Neural Graph Graph Neural Network), and the transformer variant as NG-T (Neural Graph Transformer). We refer to the appendix for more details. The code for the experiments is open-sourced[2] to facilitate reproduction of the results.

### 4.1 INR CLASSIFICATION AND STYLE EDITING

First, we evaluate our method on implicit neural representation (INR) classification and style editing, comparing against DWSNet (Navon et al., 2023) and NFN (Zhou et al., 2023a). These tasks involve exclusively the same MLP architecture per dataset.

**Setup.** For the INR classification task, we use two datasets. The first dataset contains a single INR (Sitzmann et al., 2020) for each image from the MNIST dataset (LeCun et al., 1998). The second contains one INR per image from Fashion MNIST (Xiao et al., 2017). An INR is modeled by a small MLP that learns the mapping from an input coordinate to the grayscale value of the image (or to the RGB value for colored images). Each INR is separately optimized to reconstruct its corresponding image. We use the open-source INR datasets provided by Navon et al. (2023).

In the style editing task, we assess the model's ability to predict weight updates for the INRs using the MNIST INR dataset. The objective is to enlarge the represented digit through dilation. We follow the same training objective as Zhou et al. (2023a).

**Results.** In Figures 2a and 2b, we observe that our approach outperforms the equivariant baseline by up to $+11.6\%$ on MNIST and up to $+7.8\%$ on Fashion MNIST in INR classification. In Figure 2c, we also observe performance gains over both DWSNet and NFN in the style editing task. Interestingly, the baseline can perform equally well in terms of training loss, but our graph-based approach exhibits better generalization performance. We do not report any non-equivariant baselines as they all perform considerably worse. The results highlight that increasing the number of probe features can further improve the performance. Furthermore, the probe features are effective even in a setting where we require an output per parameter as opposed to a global prediction. Despite the advancements, a performance gap persists when compared to the performance of neural networks applied directly to the original data. We hypothesize that a key factor contributing to this gap is overfitting, compounded by the absence of appropriate data augmentation techniques to mitigate it. This hypothesis is supported by the findings of Navon et al. (2023), who observed that the addition of 9 INR views as a data augmentation strategy increased accuracy by $+8\%$.

**Importance of positional embeddings.** We ablate the significance of positional embeddings in the task of MNIST INR classification. Without positional embeddings, NG-GNN achieves an accuracy of $83.9_{\pm 0.3}\%$, and NG-T attains $77.9_{\pm 0.7}\%$. This is a decrease of 7.5 and 14.5 points respectively, which highlights the importance of positional embeddings.

---

[2]https://github.com/mkofinas/neural-graphs

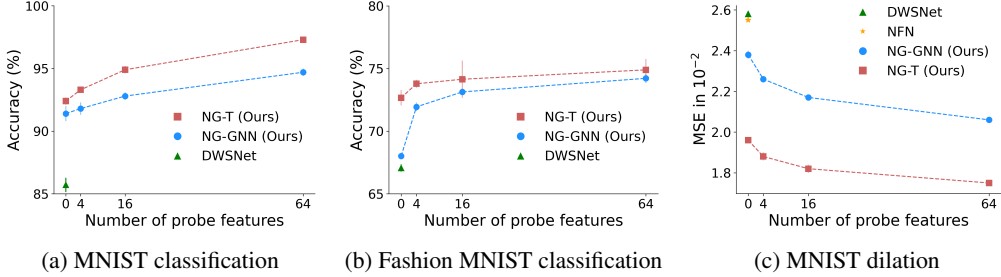

|                          |                                |                      |
| :----------------------: | :----------------------------: | :------------------: |
| (a) MNIST classification | (b) Fashion MNIST classification | (c) MNIST dilation |

Figure 2: INR classification and style editing. Results are averaged over 4 seeds. Classification results are in Figures 2a and 2b, and dilation results are in Figure 2c. Both NG-GNN and NG-T outperform DWSNet and NFN already with 0 probe features and significantly improve with more probe features.

Table 1: Predicting CNN generalization from weights. Kendall's $\tau$ on CIFAR10-GS (Unterthiner et al., 2020) and CNN Wild Park. Higher is better. We report the mean and standard deviation for 3 seeds.

| Method | CIFAR10-GS | CIFAR10 Wild Park |
| :--- | :---: | :---: |
| NFN$_{HNP}$ (Zhou et al., 2023a) | $0.934_{\pm0.001}$ | — |
| StatNN (Unterthiner et al., 2020) | $0.915_{\pm0.002}$ | $0.719_{\pm0.010}$ |
| NG-GNN (Ours) | $0.930_{\pm0.001}$ | $0.804_{\pm0.009}$ |
| NG-T (Ours) | $\mathbf{0.935_{\pm0.000}}$ | $\mathbf{0.817_{\pm0.007}}$ |

## 4.2 PREDICTING CNN GENERALIZATION FROM PARAMETERS

Next, we consider the task of predicting the generalization performance of a CNN image classifier based on its parameters (Unterthiner et al., 2020; Eilertsen et al., 2020; Schürholt et al., 2022b).

**Setup.** We consider two datasets with increasing complexity. First, we use the CIFAR10-GS dataset from the *Small CNN Zoo* (Unterthiner et al., 2020), where GS denotes that images are converted to grayscale. We closely follow the CNN generalization setting from Zhou et al. (2023a); we use Kendall's correlation $\tau$ to measure the performance, and we explore both MSE and binary cross entropy as loss functions. We compare our method with NFN (Zhou et al., 2023a) and StatNN (Unterthiner et al., 2020), a method that computes statistics (mean, standard deviation, quantiles) for weights and biases of each layer, concatenates them, and processes them with an MLP.

The Small CNN Zoo comprises identical networks, *i.e.*, all CNNs have the exact same convolutional layers, non-linearities, and linear layers, and only vary in their weights and biases, resulting from different initialization and training hyperparameters. Our method can in principle be applied to datasets that contain neural networks with varying architectures, but we have yet to see whether a single model can learn a joint representation. To test this, we introduce a new dataset of CNNs, which we term *CNN Wild Park*. The dataset consists of 117,241 checkpoints from 2,800 CNNs, trained for up to 1,000 epochs on CIFAR10. The CNNs vary in the number of layers, kernel sizes, activation functions, and residual connections between arbitrary layers. We describe the dataset in detail in Appendix E.1. We compare our method against StatNN (Unterthiner et al., 2020). NFN is inapplicable in this setting, due to the heterogeneous architectures present in the dataset. Similar to the CIFAR10-GS, we use Kendall's $\tau$ to as an evaluation metric.

**Results.** We report results in Table 1. In both settings, our NG-T outperforms the baselines, while NG-N performs slightly worse than NG-T. Although the performance gap in CIFAR10-GS is narrow, on the Wild Park dataset we see a significant performance gap between StatNN and our neural graph based approaches. Here the architecture of the CNN classifier offers crucial information and the parameters alone appear to be insufficient.

**Importance of non-linearity features.** We ablate the significance of non-linearity features on CNN Wild Park. Without non-linearity features, NG-GNN achieves $0.778_{\pm0.018}$, and NG-T $0.728_{\pm0.010}$, resulting in a performance decrease by $2.6$ and $8.9$, respectively. This highlights that including the non-linearities as features is crucial, especially for the Transformer.

Table 2: Learning to optimize. Test image classification accuracy (%) after 1,000 steps. We repeat the evaluation for 5 random seeds and report the mean and standard deviation.

| Optimizer | FashionMNIST (validation task) | CIFAR-10 (test task) |
|---|---|---|
| Adam (Kingma & Ba, 2015) | $80.97_{\pm 0.66}$ | $54.76_{\pm 2.82}$ |
| FF (Metz et al., 2019) | $85.08_{\pm 0.14}$ | $57.55_{\pm 1.06}$ |
| LSTM (Metz et al., 2020) | $85.69_{\pm 0.23}$ | $59.10_{\pm 0.66}$ |
| NFN (Zhou et al., 2023a) | $83.78_{\pm 0.58}$ | $57.95_{\pm 0.64}$ |
| NG-GNN (Ours) | $85.91_{\pm 0.37}$ | $\mathbf{64.37_{\pm 0.34}}$ |
| NG-T (Ours) | $\mathbf{86.52_{\pm 0.19}}$ | $60.79_{\pm 0.51}$ |

## 4.3 Learning to optimize

A novel application where leveraging the graph structure of a neural network may be important, yet underexplored, is "learning to optimize" (L2O) (Chen et al., 2022; Amos, 2022). L2O's task is to train a neural network (*optimizer*) that can optimize the weights of other neural networks (*optimizee*) with the potential to outperform hand-designed gradient-descent optimization algorithms (SGD, Adam (Kingma & Ba, 2015), etc.). Original L2O models were based on recurrent neural networks (Andrychowicz et al., 2016; Wichrowska et al., 2017; Chen et al., 2020). Recent L2O models are based on simple and more efficient MLPs, but with stronger momentum-based features (Metz et al., 2019; 2020; Harrison et al., 2022; Metz et al., 2022a).

**Setup.** We implement two strong learnable optimizer baselines: per-parameter feed forward neural network (FF) (Metz et al., 2019) and its stronger variant (FF + layerwise LSTM, or just LSTM) (Metz et al., 2020). FF predicts parameter updates for each parameter ignoring the structure of an optimizee neural network. LSTM adds layer features (average features of all the weights within the layer) with the LSTM propagating these features between layers. We also combine FF with NFN (Zhou et al., 2023a) as another baseline, and add Adam as a non-learnable optimizer baseline. We compare these baselines to FF combined with our NG-GNN or Relational transformer (NG-T). The combinations of FF with the NFN, NG-GNN, and NG-T models are obtained analogously to LSTM, *i.e.*, the weight features are first transformed by these models and then passed to FF to predict parameter updates.

In all learnable optimizers, given an optimizee neural network with weights $\{\mathbf{W}^{(l)}\}_{l=1,...,L}$, the features include weight gradients $\{\nabla \mathbf{W}^{(l)}\}_{l=1,...,L}$ and momentums at five scales: $[0.5, 0.9, 0.99, 0.999, 0.9999]$ following Metz et al. (2019). These features are further preprocessed with the log and sign functions (Andrychowicz et al., 2016), so the total number of node and edge features is $d_{\mathbf{V}} = d_{\mathbf{E}} = 14(w_{\max} \cdot h_{\max})$ in our experiments, where $w_{\max} \cdot h_{\max}$ comes from flattening the convolutional kernels according to Section 2.2.

We follow a standard L2O setup and train each optimization method on FashionMNIST (Xiao et al., 2017) followed by evaluation on FashionMNIST as well as on CIFAR-10 (Krizhevsky et al., 2009). For FashionMNIST, we use a small three layer CNN with 16, 32, and 32 channels, and $3 \times 3$ ($w_{\max} \times h_{\max}$) kernels in each layer followed by global average pooling and a classification layer. For CIFAR-10, we use a larger CNN with 32, 64, and 64 channels.

In order to perform well on CIFAR-10, the learned optimizer has to generalize to both a larger architecture and a different task. During the training of the optimizers, we recursively unroll the optimizer for 100 inner steps in each outer step and train for up to 1,000 outer steps in total. Once the optimizer is trained, we apply it for 1,000 steps on a given task. We train all the learnable optimizers with different hyperparameters (hidden size, number of layers, learning rate, weight decay, number of outer steps) and choose the configuration that performs the best on FashionMNIST. This way, CIFAR-10 is a test task that is not used for the model/hyperparameter selection in our experiments.

**Results.** On the validation task (FashionMNIST), the best performing method is NG-T followed by NG-GNN (Table 2). The best performing baseline is LSTM followed by FF. On the test task (CIFAR-10), NG-GNN outperforms NG-T indicating its strong generalization capabilities. LSTM is again the best performing baseline followed by NFN. The training loss and test accuracy curves on CIFAR-10 also reveal that NG-GNN's gain over other methods gradually grows demonstrating its great potential in the L2O area (Figure 3). We believe that NG-T may also generalize better to new tasks once it is trained on more tasks, following recent trends in L2O (Metz et al., 2022b).

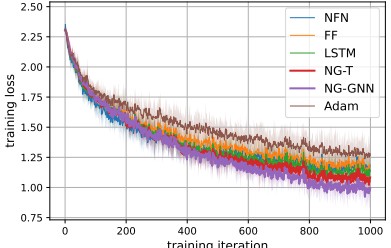 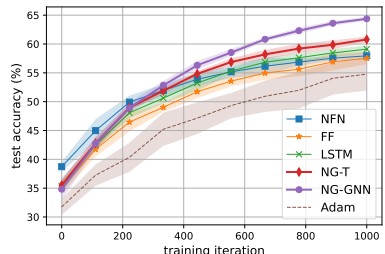

Figure 3: Training (left) and testing (right) curves on CIFAR-10 for the baseline optimizers (Adam, FF, LSTM, NFN) and the optimizers trained with our NG-GNN and NG-T. NG-GNN and NG-T outperform the baselines with the NG-GNN performing the best on this task.

## 5 RELATED WORK

**Networks for networks.** A recent line of work studies how to learn representations for trained classifiers (Baker et al., 2017; Eilertsen et al., 2020; Unterthiner et al., 2020; Schürholt et al., 2021; 2022b;a) to predict their generalization performance, or other properties that provide insight into neural networks. Dupont et al. (2022); De Luigi et al. (2023) learn low-dimensional encodings for INRs for generation or other downstream tasks. These works either flatten the network parameters or compute parameter statistics and process them with a standard MLP. Using statistics respects the symmetries of the neural network, but discards fine-grained details that can be important for the task. On the other hand, architectures that flatten the network parameters are not equivariant with respect to the symmetries of the neural network. This is undesirable because models that are functionally identical – obtained by permuting the weights and biases appropriately – can receive vastly different predictions. Schürholt et al. (2021) proposed neuron permutation augmentations to address the symmetry problem, however such an approach can require vastly more training compute due to the size of the permutation group: $\prod_l d_l!$. Other related works (Peebles et al., 2022; Ashkenazi et al., 2023; Knyazev et al., 2021; Erkoç et al., 2023) encode and/or decode neural network parameters mainly for reconstruction and generation purposes. However, similarly to the aforementioned works, these do not address the symmetry problem in a principled way.

**The symmetry problem.** Three recent studies address these shortcomings and propose equivariant linear layers that achieve equivariance through intricate weight-sharing patterns (Navon et al., 2023; Zhou et al., 2023a;b). They compose these layers to create both invariant and equivariant networks for networks. So far, their performance on INR classification shows that there is still a gap when compared to image classification using CNNs. Similar to these studies, our proposed method maintains the symmetry of the neural network. In contrast to these works, by integrating the graph structure in our neural network, we are no longer limited to homogeneous architectures and can process *heterogeneous* architectures for a much wider gamut of applications.

## 6 CONCLUSION AND FUTURE WORK

We have presented an effective method for processing neural networks with neural networks by representing the input neural networks as neural graphs. Our experiments showcase the breadth of applications that this method can be applied to. The general framework is flexible enough so that domain-specific adaptations, *e.g.*, including gradients as inputs, are simple to add. Furthermore, by directly using the graph, we open the door for a variety of applications that require processing varying architectures, as well as new benchmarks for graph neural networks.

**Limitations.** While our method is versatile enough to handle neural networks with a diverse range of architectural designs, the scope of our investigation was limited to two different architecture families (MLPs and CNNs), and only theoretically demonstrated how to represent transformers as neural graphs. Another limitation is that our method's strong performance on INRs is confined to 2D images, which restricts its applicability. Extending our approach to handle neural radiance fields (Mildenhall et al., 2020) would substantially broaden its utility.

## ACKNOWLEDGMENTS

We sincerely thank Siamak Ravanbakhsh for the invaluable guidance on the equivariance discussion in Appendix A. Furthermore, we thank SURF (`www.surf.nl`) for the support in using the National Supercomputer Snellius. The experiments were also enabled in part by computational resources provided by Calcul Quebec and Compute Canada.

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

# A  EQUIVARIANCE PROPERTIES

In this section, we will address three points. First, we state the symmetry group of the neural graph explicitly. Then, we restate the symmetry group of prior works and establish that any model that is equivariant to neural graphs will automatically be equivariant to the symmetry of the prior work. Lastly, we discuss why we believe that our choice of symmetry is usually more appropriate for processing neural network parameters as input.

## A.1  NEURAL GRAPH SYMMETRIES

Since we represent the input as a graph, we can use many existing properties of graphs. Namely, it is well-established that the symmetries of graphs are given by the symmetric group $S_n$ of order $n = \sum_{l=0}^{L} d_l$ (the total number of nodes). Let $\rho$ denote the group representation that maps a permutation $\pi \in S_n$ to the corresponding permutation matrix. A group action $\alpha$ on the graph $\mathcal{G} = (\boldsymbol{V}, \boldsymbol{E})$ can be defined as:

$$\alpha(\pi, (\boldsymbol{V}, \boldsymbol{E})) = \left(\rho(\pi)\boldsymbol{V}, \ \rho(\pi)^\top \boldsymbol{E}\rho(\pi)\right). \tag{6}$$

We want to process the neural graph with a neural network that maintains the equivariance with respect to the specified symmetries. One such example is the standard message passing graph neural network (MPNN), which is equivariant (*cf.* Bronstein et al. (2021)) since all individual operations are designed to be equivariant.

## A.2  NEURON PERMUTATION GROUP

Similar to Zhou et al. (2023a), we focus on the neuron permutation (NP) group to simplify the exposition. The main difference to the symmetry usually present in neural network parameters[3] is that input and output neurons can be freely permuted as well. As we describe in Section 2.4, we use positional embeddings rather than modifying the architecture to disambiguate the ordering of input and output neurons, so this difference is not directly relevant to the following discussion.

We now restate the definitions of Zhou et al. (2023a) in our notation to allow for discussion of the similarities and differences. First, the NP group is defined as $\mathcal{S} = S_{d_0} \times \ldots \times S_{d_L}$. The elements of the group are then $\pi' = (\pi_0, \ldots, \pi_L)$. For a specific layer $l$, the actions $\beta$ are defined as

$$\beta\left(\pi', \left(\mathbf{W}^{(l)}, \mathbf{b}^{(l)}\right)\right) = \left(\rho(\pi_l)\mathbf{b}^{(l)}, \ \rho(\pi_l)\mathbf{W}^{(l)}\rho(\pi_{l-1})^\top\right). \tag{7}$$

We can already see the similarity between Equations (6) and (7). To make this connection more concrete, notice that $\mathcal{S}$ is a subgroup of $S_n$. This is because $\mathcal{S}$ is defined as a direct product of symmetric groups whose degrees sum up to $n$ by definition. We can also verify that any action of $\mathcal{S}$ corresponds to an action of $S_n$. To see this, define the group representation $\rho'$ of $\mathcal{S}$ as the block permutation matrix:

$$\rho'(\pi') = \begin{bmatrix} \rho(\pi_0) & \mathbf{0} & \ldots & \mathbf{0} \\ \mathbf{0} & \rho(\pi_1) & \ddots & \vdots \\ \vdots & \ddots & \ddots & \mathbf{0} \\ \mathbf{0} & \ldots & \mathbf{0} & \rho(\pi_L) \end{bmatrix} \tag{8}$$

Then, we can rewrite $\beta$ as $\beta'$ in exactly the same form as $\alpha$:

$$\beta'(\pi', (\boldsymbol{V}, \boldsymbol{E})) = \left(\rho'(\pi')\boldsymbol{V}, \ \rho'(\pi')^\top \boldsymbol{E}\rho'(\pi')\right). \tag{9}$$

Since $\mathcal{S}$ is a subgroup of $S_n$, **any model that is $S_n$-equivariant must also be $\mathcal{S}$-equivariant**. In other words, choosing models like MPNNs with our neural graph ensures that we satisfy the permutation symmetries of neural networks established in the literature.

---

[3]Defined by Zhou et al. (2023a) as the hidden neuron permutation (HNP) group

## A.3 DISCUSSION

Why would we prefer one symmetry group over the other? The main factor is that $S_n$ encompasses many choices of $\mathcal{S}$ at once. Using a single $S_n$-equivariant model, our method is automatically equivariant to *any* choice of $d_0, \ldots, d_L$ that sums to $n$; the same model can process many different types of architectures. In contrast, prior works (Navon et al., 2023; Zhou et al., 2023a) propose models that are solely equivariant to $\mathcal{S}$, which restrict the models to a specific choice for $d_0, \ldots, d_L$ at a time; hence, it is not possible to process different network architectures with DWSNet (Navon et al., 2023) and NFN (Zhou et al., 2023b). Thus, when there is a need to process different architectures with the same model, the neural graph representation is more suitable.

Another aspect to consider is that DWSNet and NFN learn different parameters for each layer in the input neural network. In contrast, a neural network operating on the neural graph shares parameters for the update function that it applies at every node, irrespective of the layer it belongs to. This difference is particularly relevant in scenarios where performance on a single, fixed architecture is the primary focus. Under such conditions, the inherent equivariance of our model may be overly constraining, failing to meet the criteria for minimal equivariance as outlined by Zhou et al. (2023b). This has two potential effects: the reduced number of parameters could force the model to learn more generalizable features that work across layers, or it could simply result in not having enough parameters to fit the data. We remedy this shortcoming by integrating layer embeddings that identify the layer each node belongs to. These features are added to the node representations and enable graph neural networks to differentiate each node depending on the layer.

Experimentally, we observe that our model not only outperforms the baselines but also generalizes better. In Figure 4, we plot the train and test losses for MNIST INR classification. We observe that even at points where the train losses match, the test loss of NG-T is lower. The results confirm that the parameter sharing from $S_n$ equivariance not only does not harm the performance but might even benefit generalization.

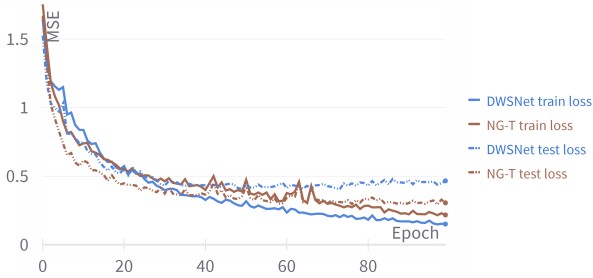

Figure 4: Generalization comparison between DWSNet and NG-T on MNIST INR classification. We observe better generalization for NG-T compared to DWSNet

## B EXPRESSIVITY OF GNNS FOR NEURAL GRAPHS

The ability to approximate the forward pass of an input neural network can indicate the expressiveness of a neural network operating on other neural networks as inputs (Navon et al., 2023). We demonstrate that a general message-passing neural network with $L$ layers can simulate the forward pass of any input MLP with $L$ layers. We first place the input for which we intend to compute the forward pass in the node features of the input nodes in the neural graph. Then we simulate the $l$-th layer's forward pass with the $l$-th layer in the MPNN. To illustrate how this can be achieved, we start by decomposing the first layer of an MLP with activation function $\sigma$ into scalar operations:

$$f(\mathbf{x})_i = \sigma(\mathbf{W}_i \mathbf{x} + \mathbf{b}_i) = \sigma\left(\mathbf{b}_i + \sum_j \mathbf{W}_{ij} \mathbf{x}_j\right). \tag{10}$$

We compare this to the update operation for the node features $\mathbf{v}_i \to \mathbf{v}'_i$ in an MPNN:

$$\mathbf{v}'_i = \phi_u \left( \mathbf{v}_i, \sum_j \phi_m(\mathbf{v}_i, \mathbf{e}_{ij}, \mathbf{v}_j) \right). \tag{11}$$

We can draw parallels between the operations in the MPNN and how the input MLP computes the subsequent layer's activations. For improved clarity, we assume in the following that the indices $i, j$ match between the parameters and the graph's nodes and edges. A more precise version would need to add an offset $d_0$ to the index $i$ in the graph. First, we recall that the edge features $\mathbf{e}_{ij}$ contain the weights of the MLP $\mathbf{W}_{ij}$, the node features $\mathbf{v}_j$ contain the input features $\mathbf{x}_j$ (for which we compute the forward pass), and the node features $\mathbf{v}_i$ contain the biases $\mathbf{b}_i$. The first layer of the MPNN can approximate $f(\mathbf{x})_i$, by approximating the scalar product $\mathbf{W}_{ij}\mathbf{x}_j$ with the neural network $\phi_m$. Given these inputs, $\phi_m$ merely needs to approximate a multiplication between the last two inputs. The addition of the bias, followed by the activation function $\sigma$ can easily be approximated with the node update MLP $\phi_u$.

For the remaining layers in the MLP, the $\phi_u$ merely needs to ensure that it preserves the input node features until it requires them for approximating the $l$-th MLP layer in the $l$-th MPNN layer. For this, we simply include a layer embedding as a layer position indicator in the input node features. Finally, the output of the simulated forward pass coincides with the node features of the final MPNN layer.

This construction approximates the forward pass of an input neural network by using a general MPNN that operates on the neural graph. The algorithmic alignment between the operations of the MPNN and the computational steps of the forward pass indicates that the forward pass is not only possible to express, but it can also be learned efficiently (Xu et al., 2020).

## C  NEURAL GRAPH REPRESENTATIONS

### C.1  MLPS AS GRAPHS

The neural graph specifies the computation of the neural network. Edges apply the function $y_{ij} = w_{ij}x_i$ where $w_{ij}$ is the weight associated with the edge $e_{ij}$ and the input $x_i$ comes from the tail node $v_i$. The node $v_j$ applies the sum operation $x_j = b_j + \sum_i y_{ij}$ on all incoming edges' $y_{ij}$.

### C.2  CNNS AS GRAPHS

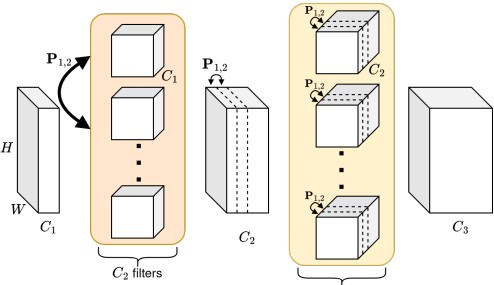

Figure 5: Permutation symmetries in a CNN, shown in a 2-layer CNN sub-network. The first convolutional layer has a filter bank of $C_2$ filters, each with $C_1$ channels, and the second layer has $C_3$ filters, each with $C_2$ channels. Permuting the filters in the first layer results in permuting the channels of the feature map. Applying a permutation in the channels of the filters in the second layer cancels out the permutations, resulting in permutation invariance.

**Flattening layer**   We propose two options for dealing with the flattening layer of traditional CNN architectures: adding one extra (virtual) layer, or repeating nodes in the last convolutional layer. For either option, we are binding the neural graph to a specific spatial resolution, and thus, we need to encode that in the graph structure as well.

In the first option, we repeat every node in the last convolutional layer by the number of spatial locations. As an example, if the CNN produces a feature map that has spatial dimensions of $7 \times 7$ at that layer, then we would need to copy the nodes 49 times. When we copy a node, we also need to copy the edges that point to it. The weights connecting these nodes to the subsequent linear layer do not require any changes, since they already account for the spatial structure. Finally, we add positional embeddings to the copied nodes to indicate their different locations in space.

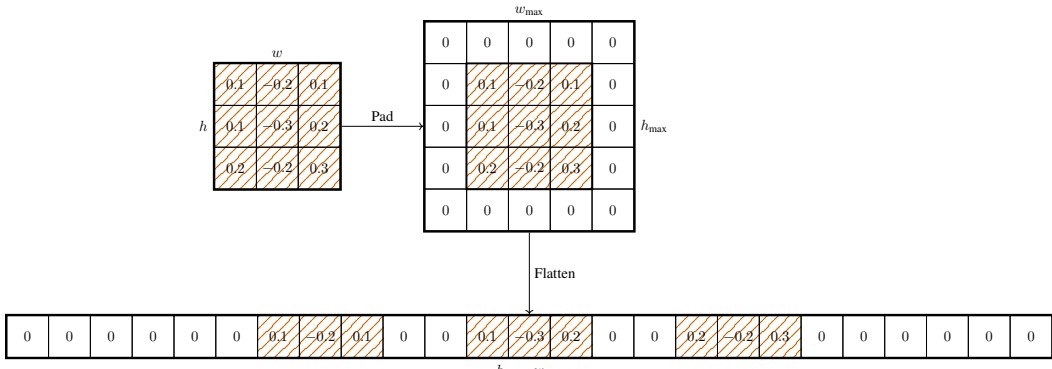

Figure 6: Representation of a convolutional kernel. We zero-pad to a predetermined maximum kernel size ($5 \times 5$ in this example) and flatten it.

An alternative option is to add an extra (virtual) layer, between the last convolutional layer and the subsequent linear layer. In that layer, we generate as many nodes as the spatial resolution of the feature map (*e.g.* 49 in the example above), for each node in the last convolutional layer. Each node in the last convolutional layer is only connected to its own set of virtual nodes, with a weight of 1. The new virtual nodes don't have any node features.

### C.3 NORMALIZATION LAYERS AS GRAPHS

Normalization layers, e.g. BatchNorm (Ioffe & Szegedy, 2015) and LayerNorm (Ba et al., 2016) are formulated as $\mathbf{y} = \gamma \odot \mathbf{x} + \beta$, which can be rewritten as a linear layer with diagonal weights $\mathbf{y} = \mathrm{diag}(\gamma)\mathbf{x} + \beta$. As such, we can treat normalization layers like linear layers: given $d$ nodes that represent the $d$-dimensional input to the normalization layer $\mathbf{x}$, we add an additional $d$ nodes that correspond to the $d$-dimensional output $\mathbf{y}$. The additional node features capture the additive terms $\beta$, while the edge features capture the multiplicative terms $\gamma$. We only add edges from $\mathbf{x}_i$ to the corresponding $\mathbf{y}_i$ to model the element-wise multiplication.

### C.4 TRANSFORMERS AS GRAPHS

Multi-head self-attention layers initially apply linear projections to the inputs $\mathbf{X}$. In total, assuming H heads, we have $3H$ linear layers applied independently to the inputs.

$$\mathbf{Q}_h = \mathbf{X}\mathbf{W}_h^Q \tag{12}$$

$$\mathbf{K}_h = \mathbf{X}\mathbf{W}_h^K \tag{13}$$

$$\mathbf{V}_h = \mathbf{X}\mathbf{W}_h^V, \tag{14}$$

with $h \in \{1, \ldots, H\}$. Each head is followed by a dot-product attention layer $\mathbf{Y}_h = \mathrm{s}(\mathbf{Q}_h\mathbf{K}_h^T)\mathbf{V}_h$, where s is the softmax function. Finally, we concatenate all heads and perform a final linear projection:

$$\mathrm{MHSA}(\mathbf{X}) = \mathrm{Concat}(\mathbf{Y}_1, \ldots, \mathbf{Y}_H)\mathbf{W}^O. \tag{15}$$

A multi-head self attention layer takes $d$-dimensional vectors as inputs and produces $d_H$-dimensional vectors as output of each head, which are then concatenated and linearly projected to $d$ dimensions. In the neural graph construction, we add $d$ nodes for each dimension of the input, $H \cdot d_H$ nodes for each dimension of each head, and $d$ nodes for each dimension of the output.

We model the 3 different types of linear projections with multidimensional edge features. More specifically, for each edge feature we have $\mathbf{e}_{ij}^h = \left( \left(\mathbf{W}_h^Q\right)_{ij}, \left(\mathbf{W}_h^K\right)_{ij}, \left(\mathbf{W}_h^V\right)_{ij} \right)$. Since dot-product attention is a parameter-free operation, we don't model explicitly and let the neural graph network approximate it. The concatenation of all heads is automatically handled by the neural graph itself by connecting the appropriate nodes from each head through the output weights $\mathbf{W}^O$ to the corresponding output node. The final projection $\mathbf{W}^O$ is treated as a standard linear layer.

## D    EXPERIMENTAL SETUP DETAILS

We provide the code for our experiments in the supplementary material. Here, we list the hyperparameters used in our experiments.

### D.1    INR CLASSIFICATION AND STYLE EDITING

The NG-T model consists of 4 layers with 64 hidden dimensions, 4 attention heads, and dropout at 0.2 probability. In the classification setting, we concatenate the node representations that belong to the final layer of the INR and insert this into a classification head. Together with the classification head, this results in 378,090 trainable parameters.

The NG-GNN model consists of 4 layers, 64 hidden dimensions, and applies dropout at 0.2 probability. In the classification setting, we concatenate the node representations that belong to the final layer of the INR and insert this into a classification head. Together with the classification head, this results in 348,746 trainable parameters.

We use the same architectures without the final aggregation and the classification head in the style editing experiment. The NG-GNN additionally uses the reversed edge features.

We train for 100 epochs on MNIST and for 150 epochs on FMNIST. We apply the same early stopping protocol as Navon et al. (2023).

### D.2    PREDICTING CNN GENERALIZATION FROM PARAMETERS

We do a small hyperparameter sweep for all models. For the StatNN baseline, we use 1000 hidden dimensions resulting in a model resulting in 1,087,001 trainable parameters.

For the NG-T we use 4 layers with 8 hidden dimensions for the edges and 16 for the nodes, 1 attention head, and no dropout. We concatenate the node representations of the final layer and insert them into the classification head.

For the NG-GNN model, we use 64 hidden dimensions, resulting in 369,025 trainable parameters. We concatenate the node representations of the final layer and insert them into the classification head.

We train all models for 10 epochs.

### D.3    PROBE FEATURES

We use probe features in the INR classification and INR style editing experiments. We do not use probe features in the task of predicting CNN generalization, or the learning to optimize task.

## E    DATASET DETAILS

### E.1    SMALL CNN WILD PARK

We construct the CNN Wild Park dataset by training 2800 small CNNs with different architectures for 200 to 1000 epochs on CIFAR10. We retain a checkpoint of its parameters every 10 steps and also record the test accuracy. The CNNs vary by:

- Number of layers $L \in [2, 3, 4, 5]$ (note that this does not count the input layer).
- Number of channels per layer $c_l \in [4, 8, 16, 32]$.
- Kernel size of each convolution $k_l \in [3, 5, 7]$.
- Activation functions at each layer are one of ReLU, GeLU, tanh, sigmoid, leaky ReLU, or the identity function.
- Skip connections between two layers with at least one layer in between. Each layer can have at most one incoming skip connection. We allow for skip connections even in the case when the number of channels differ, to increase the variety of architectures and ensure independence between different architectural choices. We enable this by adding the skip connection only to the $min(c_n, c_m)$ nodes.

We divide the dataset into train/val/test splits such that checkpoints from the same run are **not** contained in both the train and test splits.

## F EXPERIMENT RESULTS

We provide the exact results of our experiments for reference and easier comparison.

Table 3: Classification of MNIST INRs. All graph-based models outperform the baselines.

| Model | # probe features | Accuracy in % |
|---|---|---|
| MLP (Navon et al., 2023) | — | $17.6_{\pm 0.0}$ |
| Set NN (Navon et al., 2023) | — | $23.7_{\pm 0.1}$ |
| DWSNet (Navon et al., 2023) | — | $85.7_{\pm 0.6}$ |
| NG-GNN (Ours) | 0 | $91.4_{\pm 0.6}$ |
| NG-GNN (Ours) | 4 | $91.8_{\pm 0.5}$ |
| NG-GNN (Ours) | 16 | $92.8_{\pm 0.3}$ |
| NG-GNN (Ours) | 64 | $94.7_{\pm 0.3}$ |
| NG-T (Ours) | 0 | $92.4_{\pm 0.3}$ |
| NG-T (Ours) | 4 | $93.3_{\pm 0.2}$ |
| NG-T (Ours) | 16 | $94.9_{\pm 0.3}$ |
| NG-T (Ours) | 64 | $\mathbf{97.3_{\pm 0.2}}$ |

Table 4: Classification of Fashion MNIST INRs. All graph-based models outperform the baselines.

| Model | # probe features | Accuracy in % |
|---|---|---|
| MLP (Navon et al., 2023) | — | $19.9_{\pm 0.5}$ |
| Set NN (Navon et al., 2023) | — | $22.3_{\pm 0.4}$ |
| DWSNet (Navon et al., 2023) | — | $67.1_{\pm 0.3}$ |
| NG-GNN (Ours) | 0 | $68.0_{\pm 0.2}$ |
| NG-GNN (Ours) | 4 | $71.9_{\pm 0.3}$ |
| NG-GNN (Ours) | 16 | $73.1_{\pm 0.3}$ |
| NG-GNN (Ours) | 64 | $74.2_{\pm 0.4}$ |
| NG-T (Ours) | 0 | $72.7_{\pm 0.6}$ |
| NG-T (Ours) | 4 | $73.8_{\pm 0.3}$ |
| NG-T (Ours) | 16 | $74.1_{\pm 1.5}$ |
| NG-T (Ours) | 64 | $\mathbf{74.8_{\pm 0.9}}$ |

Table 5: Dilating MNIST INRs. Mean-squared error (MSE) computed between the reconstructed image and dilated ground-truth image. Lower is better.

| Model | # probe features | MSE in $10^{-2}$ |
|---|---|---|
| DWSNet (Navon et al., 2023) | — | $2.58_{\pm 0.00}$ |
| NFN (Zhou et al., 2023a) | — | $2.55_{\pm 0.00}$ |
| NG-GNN (Ours) | 0 | $2.38_{\pm 0.02}$ |
| NG-GNN (Ours) | 4 | $2.26_{\pm 0.01}$ |
| NG-GNN (Ours) | 16 | $2.17_{\pm 0.01}$ |
| NG-GNN (Ours) | 64 | $2.06_{\pm 0.01}$ |
| NG-T (Ours) | 0 | $1.96_{\pm 0.00}$ |
| NG-T (Ours) | 4 | $1.88_{\pm 0.02}$ |
| NG-T (Ours) | 16 | $1.82_{\pm 0.02}$ |
| NG-T (Ours) | 64 | $\mathbf{1.75_{\pm 0.01}}$ |

Table 6: Ablation study on the importance of non-linearity embeddings in the task of predicting CNN generalization from weights. Performance is measured as Kendall's $\tau$ on CNN Wild Park. Higher is better. Removing the non-linearity embeddings results in performance decrease by 2.6 for NG-GNN and 8.9 for NG-T. The ablation highlights that including the non-linearities as features in the neural graph is crucial, especially for the Transformer variant.

| Method | Kendall's $\tau$ |
|---|---|
| StatNN (Unterthiner et al., 2020) | $0.719_{\pm 0.010}$ |
| NG-GNN (Ours) | $\mathbf{0.804}_{\pm \mathbf{0.009}}$ |
| NG-GNN w/o activation embedding | $0.778_{\pm 0.018}$ |
| NG-T (Ours) | $\mathbf{0.817}_{\pm \mathbf{0.007}}$ |
| NG-T w/o activation embedding | $0.728_{\pm 0.010}$ |

Table 7: Ablation study on the importance of positional embeddings in the task of MNIST INR classification. Removing positional embeddings results in a decrease of 7.5 points for NG-GNN and 14.5 for NG-T, which highlights the importance of positional embeddings.

| Method | Accuracy in $\%$ |
|---|---|
| NG-GNN (Ours) | $\mathbf{91.4}_{\pm \mathbf{0.6}}$ |
| NG-GNN w/o positional embeddings | $83.9_{\pm 0.3}$ |
| NG-T (Ours) | $\mathbf{92.4}_{\pm \mathbf{0.3}}$ |
| NG-T w/o positional embeddings | $77.9_{\pm 0.7}$ |

