# OpenReview forum: "Graph Neural Networks for Learning Equivariant Representations of Neural Networks"
_ICLR.cc/2024/Conference — ICLR 2024 oral_

### Official Review · Reviewer_qkCh · 2023-10-31

**Soundness:** 3 good
**Presentation:** 3 good
**Contribution:** 3 good
**Rating:** 8
**Confidence:** 4

**Summary:**

The paper considers the task of creating neural networks that take other neural networks as input. This problem has been considered in the machine learning literature which often does not take into account the different symmetries of neural network parameters, or only considers simple and fixed architectures such as multi-layer perceptrons or convolutional neural networks (Navon et al., 2023; Zhou et al., 2023a). The paper extends those works and proposes a more flexible approach where different architectures of neural networks can be given as input their neural network (i.e. different number of layers, widths, non-linearities). The paper also considers other types of layers such as residual connections and normalizations that take respect permutation equivariance across the different layers.

**Strengths:**

In general, the paper is well motivated and easy to understand. The paper also shows strong improvement over baselines in three different tasks which are: INR classification of 2d images and style editing, predicting CNN performance and learning optimizers. In the supplementary material, an ablation study of the proposed probe features (that consider representations at different layers of the neural networks) is provided. I think that the paper is relevant to the machine learning community. It is a stepping stone to more general neural networks that take other neural networks as input.

**Weaknesses:**

Despite considering new types of layers (e.g. residual connections) compared to the literature, other types of layers such as multi-head attention layers are still missing. Moreover, only the MLP and convolutional layers have been tested in the experiments. The paper could be improved a lot if it experimentally showed that the other types of proposed layers can also help improve the performance of their model.
The paper also does not discuss the expressive power of their approach compared to the baselines, nor how difficult it is to scale their approach to very large neural networks/graphs.

**Questions:**

Assuming that the architecture of neural networks is fixed during training and test, how scalable is the proposed approach compared to the baselines (Navon et al., 2023; Zhou et al., 2023a)? In particular, can the proposed approach consider the same size of neural networks as the baselines for a fixed maximum memory allocation? Or can the proposed approach consider even larger architectures than the baselines?

Is the expressive power of the proposed approach the same as the baselines?

---

> ### Author Response · Authors · 2023-11-23
> **Author response to Reviewer qkCh (1)**
>
> We would like to thank the reviewer for their time and insightful comments.
>
> > Despite considering new types of layers (e.g. residual connections) compared to the literature, other types of layers such as multi-head attention layers are still missing. Moreover, only the MLP and convolutional layers have been tested in the experiments. The paper could be improved a lot if it experimentally showed that the other types of proposed layers can also help improve the performance of their model.
>
> Thank you for the suggestion. We provide recipes for normalization layers and multi-head self-attention layers in the updated manuscript in appendix C. For completeness, we also describe the two types of building blocks here.
>
> Normalization layers, e.g. BatchNorm [3] and LayerNorm [4] are formulated as $\mathbf{y} = \gamma \odot \mathbf{x} + \beta$, which can be rewritten as a linear layer with diagonal weights $\mathbf{y} = \mathrm{diag}(\gamma) \mathbf{x} + \beta$. As such, we can treat normalization layers like linear layers: given $d$ nodes that represent the $d$-dimensional input to the normalization layer $\mathbf{x}$, we add an additional $d$ nodes that correspond to the $d$-dimensional output $\mathbf{y}$. The additional node features capture the additive terms $\beta$, while the edge features capture the multiplicative terms $\gamma$. We only add edges from $\mathbf{x}_i$ to the corresponding $\mathbf{y}_i$ to model the element-wise multiplication.
>
> Multi-head self-attention layers initially apply linear projections to the inputs $\mathbf{X}$. In total, assuming H heads, we have $3H$ linear layers applied independently to the inputs.
> $$ \mathbf{Q}_h = \mathbf{X} \mathbf{W}_h^Q, \mathbf{K}_h = \mathbf{X} \mathbf{W}_h^K, \mathbf{V}_h = \mathbf{X} \mathbf{W}_h^V, h \in \{1, \ldots, H\} $$
> Each head is followed by a dot-product attention layer $\mathbf{Y}_h = \mathrm{s}(\mathbf{Q}_h \mathbf{K}_h^T) \mathbf{V}_h$, where $\mathrm{s}$ is the softmax function.
> Finally, we concatenate all heads and perform a final linear projection:
> $ \textrm{MHSA}(\mathbf{X}) = \mathrm{Concat}(\mathbf{Y}_1, \ldots, \mathbf{Y}_H) \mathbf{W}^O$
>
> A multi-head self attention layer takes $d$-dimensional vectors as inputs and produces $d_H$-dimensional vectors as output of each head, which are then concatenated and linearly projected to $d$ dimensions. In the neural graph construction, we add $d$ nodes for each dimension of the input, $H \cdot d_H$ nodes for each dimension of each head, and $d$ nodes for each dimension of the output.
> We model the 3 different types of linear projections with multidimensional edge features. More specifically, for each edge feature we have $\mathbf{e}_{ij}^h=\left(\left(\mathbf{W}_h^Q\right)\_{ij}, \left(\mathbf{W}_h^K\right)\_{ij}, \left(\mathbf{W}_h^V\right)\_{ij}\right)$. Since dot-product attention is a parameter-free operation, we don't model explicitly and allow the neural graph network to approximate it. The concatenation of all heads is automatically handled by the neural graph itself by connecting the appropriate nodes from each head through the output weights $\mathbf{W}^O$ to the corresponding output node. The final projection $\mathbf{W}^O$ is treated as a standard linear layer.
>
> > The paper also does not discuss the expressive power of their approach compared to the baselines,
> >
> > Is the expressive power of the proposed approach the same as the baselines?
>
> Thank you for the suggestion. Navon et al. [1] characterize the expressive power of DWSNet by showing its ability to approximate the forward pass of an MLP with ReLU activations. In the new Appendix B, we demonstrate that an MPNN applied to neural graphs can achieve this too.
>
> > The paper does not discuss how difficult it is to scale their approach to very large neural networks/graphs.
>
> Our neural graph scales linearly with the number of parameters in the input neural network. The complexity of the graph neural network (or Transformer) decides whether it can scale to very large neural networks as inputs. Message passing neural networks typically have linear complexity, whereas Transformer has quadratic complexity in the number of nodes - this number is typically much smaller than the number of parameters/edges. Much research is being done to apply graph neural networks and Transformers to larger inputs, and applying those advances to neural graphs is an exciting future research direction.

---

> ### Author Response · Authors · 2023-11-23
> **Author response to Reviewer qkCh (2)**
>
> > Assuming that the architecture of neural networks is fixed during training and test, how scalable is the proposed approach compared to the baselines (Navon et al., 2023; Zhou et al., 2023a)? In particular, can the proposed approach consider the same size of neural networks as the baselines for a fixed maximum memory allocation? Or can the proposed approach consider even larger architectures than the baselines?
>
> To test the memory complexity of our method and the baselines, we perform a study using 3-layer INRs with increasing hidden dimensions, and train models using them as input. We set the batch size to 1 and use models with roughly the same number of parameters. The table below shows the maximum network size for each method:
>
> Method | INR Layout |
> :--- | ---|
> NFN [2] | [2, 1024, 2048, 3]
> DWS [1] | [2, 2048, 4096, 3]
> NG-GNN (Ours) | [2, 1024, 2048, 3]
> NG-T (Ours) | [2, 512, 1024, 3]
>
> We see that our GNN is as memory efficient as NFN but less so than DWS, while the Transformer is more memory hungry due to the quadratic complexity. Note though that the specific numbers are highly implementation dependent, and we did not optimize our implementation for memory efficiency. There is a large body of literature on more efficient GNNs and Transformers, and tapping into it for more efficient neural graphs is a promising direction for future work.
>
> ### References
>
> [1] Aviv Navon, Aviv Shamsian, Idan Achituve, Ethan Fetaya, Gal Chechik, and Haggai Maron. Equivariant Architectures for Learning in Deep Weight Spaces. ICML 2023.
>
> [2] Allan Zhou, Kaien Yang, Kaylee Burns, Yiding Jiang, Samuel Sokota, J Zico Kolter, and Chelsea Finn. Permutation Equivariant Neural Functionals. NeurIPS 2023.
>
> [3] Sergey Ioffe, Christian Szegedy. Batch Normalization: Accelerating Deep Network Training by Reducing Internal Covariate Shift. ICML 2015.
>
> [4] Jimmy Lei Ba, Jamie Ryan Kiros, Geoffrey E. Hinton. Layer Normalization.

---

### Official Review · Reviewer_rtFz · 2023-10-31

**Soundness:** 4 excellent
**Presentation:** 4 excellent
**Contribution:** 3 good
**Rating:** 8
**Confidence:** 4

**Summary:**

This paper introduces a new approach to designing neural networks that process the parameters of other neural networks. This is achieved by representing the input neural network as a graph. In doing so, a graph neural network can operate on the input graph while respecting parameter permutation symmetries.

The neural network is converted to a graph by the introduction of a neural graph representation. This neural graph representation is related to the computation graph, but is more compact in some cases. Neural graph representations are designed for MLPs, CNNs, and residual networks. As part of the neural graph representation, the papers introduce novel node and edge features that improve the expressivity of the graph neural networks.

The proposed method is evaluated over a diverse set of experiments: classifying implicit neural representations, predicting network generalization, and learning to optimize. Across all settings, the proposed method significantly outperforms the evaluated baselines.

**Strengths:**

The proposed method is novel. I also agree with the authors that it is an improvement over baseline methods in that this approach can handle multiple architectures with the same model and does not require bespoke layer design.

The paper is very well written and easy to understand. I was able to follow completely and feel that adequate detail was provided for me to reproduce the results.

The empirical evaluation is thorough and conclusive. The proposed method gives a significant boost to performance over the baselines. The paper evaluates performance over several established tasks. There is a good amount of variety and error bars are included for all experiments, further establishing the consistent benefits. The authors also included full source code for their method and experiments.

I feel that the contributions of this work are significant overall. This is a nice approach to designing neural networks that process other neural networks, which alleviates some of the complexity in prior work.

**Weaknesses:**

The CNN graph construction is effective but feels quite hacky. To use the method, the user must first specify a maximum kernel size. While this is unlikely to be a problem in practice, because the kernel size of modern CNNs does not vary much, it is not obvious how to extend this to other network layers that exhibit parameter sharing. For example, attention layers share parameters over sequence length which may vary significantly more than kernel size. Moreover, other standard building blocks like normalization layers are not covered in this work and there is no clear recipe provided for designing the corresponding neural graph representations.

There is little theoretical justification for the proposed approach. Prior work proves the expressivity of their methods alongside their group equivariance properties. Neither of these are explored formally in this work.

I consider the empirical results to be quite complete, but an ablation of the various design decisions introduced would be valuable. For example, I'd like to better understand how much value the probe features, non-linearity identification, positional encoding, and other components contribute to the overall performance.

**Questions:**

- It is stated that the proposed neural graphs "ensure invariance to neuron symmetries". Are you able to outline how this might be proved?
- Related to the previous question, the authors write that "natural symmetries in the neural graphs correspond exactly to neuron permutation symmetries", is this a statement that can be formalized? It is not clear to me that this is a 1:1 correspondence for all graphs considered. However, it is stated that this can be shown (Sec 2.1).
- You observe that the baseline methods are able to perform equally well on the training loss, but fail to generalize as well (Sec 4.1). Why do you think this is? Did you explore adding regularization or similar to the baseline methods to help with generalization? I wonder if the probe features or other modifications are providing some of this benefit for the proposed method.


Minor comments:

- I thought the probe features is a very neat idea that, intuitively, adds a lot of expressive power to the proposed method.
- At the end of the introduction, it is written that the proposed method "outperforms state-of-the-art approaches by a large margin". I think perhaps this would be better quantified with some specific values.
- In Section 2.3, non-linearities are described as being added to the node features. Is this done via concatenation? And what is done when there is no activation?

---

> ### Author Response · Authors · 2023-11-23
> **Author response to Reviewer rtFz (1)**
>
> We would like to thank the reviewer for their time and insightful comments.
>
> > The CNN graph construction is effective but feels quite hacky. To use the method, the user must first specify a maximum kernel size. While this is unlikely to be a problem in practice, because the kernel size of modern CNNs does not vary much, it is not obvious how to extend this to other network layers that exhibit parameter sharing. For example, attention layers share parameters over sequence length which may vary significantly more than kernel size.
>
> Indeed, our construction requires prior knowledge on a maximum allowed kernel size. Note, however, that the maximum kernel size merely decides the input feature dimensions of the edges. On the other hand, the parameter sharing of attention layers is more comparable to how a shared convolutional kernel is applied over the different spatial regions. Hence, the graph construction for attention layers is unaffected by the variability in the sequence length, similar to how it is unaffected by the variability in the image size for CNNs.
>
> > Moreover, other standard building blocks like normalization layers are not covered in this work and there is no clear recipe provided for designing the corresponding neural graph representations.
>
> Thank you for the suggestion. We provide recipes for normalization layers and multi-head self-attention layers in the updated manuscript in appendix C. For completeness, we also describe the two types of building blocks here.
>
> Normalization layers, e.g. BatchNorm [3] and LayerNorm [4] are formulated as $\mathbf{y} = \gamma \odot \mathbf{x} + \beta$, which can be rewritten as a linear layer with diagonal weights $\mathbf{y} = \mathrm{diag}(\gamma) \mathbf{x} + \beta$. As such, we can treat normalization layers like linear layers: given $d$ nodes that represent the $d$-dimensional input to the normalization layer $\mathbf{x}$, we add an additional $d$ nodes that correspond to the $d$-dimensional output $\mathbf{y}$. The additional node features capture the additive terms $\beta$, while the edge features capture the multiplicative terms $\gamma$. We only add edges from $\mathbf{x}_i$ to the corresponding $\mathbf{y}_i$ to model the element-wise multiplication.
>
> Multi-head self-attention layers initially apply linear projections to the inputs $\mathbf{X}$. In total, assuming H heads, we have $3H$ linear layers applied independently to the inputs.
> $$ \mathbf{Q}_h = \mathbf{X} \mathbf{W}_h^Q, \mathbf{K}_h = \mathbf{X} \mathbf{W}_h^K, \mathbf{V}_h = \mathbf{X} \mathbf{W}_h^V, h \in \{1, \ldots, H\} $$
> Each head is followed by a dot-product attention layer $\mathbf{Y}_h = \mathrm{s}(\mathbf{Q}_h \mathbf{K}_h^T) \mathbf{V}_h$, where $\mathrm{s}$ is the softmax function.
> Finally, we concatenate all heads and perform a final linear projection:
> $ \textrm{MHSA}(\mathbf{X}) = \mathrm{Concat}(\mathbf{Y}_1, \ldots, \mathbf{Y}_H) \mathbf{W}^O$
>
> A multi-head self attention layer takes $d$-dimensional vectors as inputs and produces $d_H$-dimensional vectors as output of each head, which are then concatenated and linearly projected to $d$ dimensions. In the neural graph construction, we add $d$ nodes for each dimension of the input, $H \cdot d_H$ nodes for each dimension of each head, and $d$ nodes for each dimension of the output.
> We model the 3 different types of linear projections with multidimensional edge features. More specifically, for each edge feature we have $\mathbf{e}_{ij}^h=\left(\left(\mathbf{W}_h^Q\right)\_{ij}, \left(\mathbf{W}_h^K\right)\_{ij}, \left(\mathbf{W}_h^V\right)\_{ij}\right)$. Since dot-product attention is a parameter-free operation, we do not model it explicitly and let the neural graph network approximate it. The concatenation of all heads is automatically handled by the neural graph itself by connecting the appropriate nodes from each head through the output weights $\mathbf{W}^O$ to the corresponding output node. The final projection $\mathbf{W}^O$ is treated as a standard linear layer.
>
> We’ve covered a variety of popular architectural choices including convolutions, skip connections, and different activation functions, which serve as inspiration for how other neural network components can be turned into neural graphs.
>
>
> > There is little theoretical justification for the proposed approach. Prior work proves the expressivity of their methods alongside their group equivariance properties. Neither of these are explored formally in this work.
>
> Thank you for the suggestion. Navon et al. [1] proved that DWSNet can approximate the forward pass of an MLP with ReLU activations as a first step towards formally characterizing the expressivity of (equivariant) networks for networks. In the new Appendix B, we demonstrate that an MPNN applied to neural graphs can achieve this too. We further formalized the equivariance property in the new Appendix A (as also suggested by Reviewer 6cHp).

---

> ### Author Response · Authors · 2023-11-23
> **Author response to Reviewer rtFz (2)**
>
> > I consider the empirical results to be quite complete, but an ablation of the various design decisions introduced would be valuable. For example, I'd like to better understand how much value the probe features, non-linearity identification, positional encoding, and other components contribute to the overall performance.
>
> Thank you for the suggestions. We have added ablation studies on using different numbers of probe features, removing the non-linearity embeddings, and removing the position embeddings in the updated manuscript, in Sections 4.1, 4.2, and Appendix F. We observe that removing either the non-linearity embeddings or the position embeddings results in a large performance drop, which indicates their importance.
>
> > It is stated that the proposed neural graphs "ensure invariance to neuron symmetries". Are you able to outline how this might be proved?
> >
> > Related to the previous question, the authors write that "natural symmetries in the neural graphs correspond exactly to neuron permutation symmetries", is this a statement that can be formalized? It is not clear to me that this is a 1:1 correspondence for all graphs considered. However, it is stated that this can be shown (Sec 2.1).
>
> We now formalize these claims in the new Appendix A. In particular, we show that the neuron symmetry group (i.e. the symmetry in the parameter space) is a subgroup of the symmetric group (i.e. the symmetry for the neural graph). This means that for a specific neural network architecture, all its permutation symmetries in the parameter space can be expressed as permutations in the nodes of the neural graph.
>
> The above implies that any model that is equivariant with respect to the symmetric group is automatically equivariant with respect to the symmetries in the parameter space. Hence, it ensures invariance to neuron symmetries.
>
> > You observe that the baseline methods are able to perform equally well on the training loss, but fail to generalize as well (Sec 4.1). Why do you think this is? Did you explore adding regularization or similar to the baseline methods to help with generalization? I wonder if the probe features or other modifications are providing some of this benefit for the proposed method.
>
> Due to the difference in the symmetry group captured by the baselines (as we outline above), the baselines have a different parameter sharing pattern. In particular, both NFN [2] and DWSNet [1] have different parameters for different layers in the input neural network. This can make them more prone to overfitting, leading to worse generalization. We’ve added Figure 5 in Appendix A that shows that our approach achieves better test loss than the baseline, even when the train loss is the same for both. Furthermore, note that the hyperparameters of the baselines are well tuned. In particular, Navon et al. report in their appendix that DWSNet is trained using AdamW with weight decay and Batch-Normalization for improved generalization, and further applied grid search to select the optimal learning rate.
>
> > In Section 2.3, non-linearities are described as being added to the node features. Is this done via concatenation? And what is done when there is no activation?
>
> Non-linearities are represented by learnable embeddings (one for each type of non-linearity) and are added to the existing node features. No activation can be seen as another “type of non-linearity”; we use one more learnable embedding that corresponds to the identity activation.
>
> ### References
>
> [1] Aviv Navon, Aviv Shamsian, Idan Achituve, Ethan Fetaya, Gal Chechik, and Haggai Maron. Equivariant Architectures for Learning in Deep Weight Spaces. ICML 2023.
>
> [2] Allan Zhou, Kaien Yang, Kaylee Burns, Yiding Jiang, Samuel Sokota, J Zico Kolter, and Chelsea Finn. Permutation Equivariant Neural Functionals. NeurIPS 2023.
>
> [3] Sergey Ioffe, Christian Szegedy. Batch Normalization: Accelerating Deep Network Training by Reducing Internal Covariate Shift. ICML 2015.
>
> [4] Jimmy Lei Ba, Jamie Ryan Kiros, Geoffrey E. Hinton. Layer Normalization.

---

### Official Review · Reviewer_6cHp · 2023-11-01

**Soundness:** 2 fair
**Presentation:** 3 good
**Contribution:** 3 good
**Rating:** 6
**Confidence:** 4

**Summary:**

This work builds new neural networks that process other neural network parameters, by processing the computation graphs of the data neural networks. They do this processing via graph neural networks and graph transformers. Their models can then process htereogeneous architectures, as opposed to previous equivariant models. Experiments are conducted on several tasks involving processing neural networks.

**Strengths:**

1. The method can handle different nonlinearities, residual connections, and sizes of neural network.
2. The graph framework is flexible, as it allows different types of base model, e.g. the GNN and Transformer that they consider in this work.
3. Many types of empirical evidence, which shows the benefits of the method. The learned optimization experiments are particularly interesting.

**Weaknesses:**

1. While probe features improve performance, they are giving privileged information that is not quite in the same learning regime as other related works, which only take in parameters. With enough probe features, you are essentially inputting the original MNIST image into your neural network.
2. Many claims of invariance or equivariance to permutation symmetries, without any proofs.

**Questions:**

1. Why do the MNIST dilation numerical results differ so much from those in the Zhou et al. paper? They achieve about .070, whereas you achieve about .02.
2. Could you report how many probe features are used in each of your experiments?

---

> ### Author Response · Authors · 2023-11-23
> **Author response to Reviewer 6cHp**
>
> We would like to thank the reviewer for their time and insightful comments.
>
> > While probe features improve performance, they are giving privileged information that is not quite in the same learning regime as other related works, which only take in parameters. With enough probe features, you are essentially inputting the original MNIST image into your neural network.
>
> While the reviewer raises a valid point indeed, we would like to clarify that the results for INR classification and style editing in Figure 3 show that our model improves over the baselines even without probe features (i.e., number of probe features = 0). We further observe that using only 4 probe features (less than 1% of the number of pixels) already leads to significant improvements. As an example, in Fashion MNIST INR classification, we have an absolute accuracy gain of 3.9% for our GNN, and 1.1% for the Transformer. We highlight these results more in the revised text.
>
> > Many claims of invariance or equivariance to permutation symmetries, without any proofs.
>
> Thank you for the suggestion. We now formalize these claims in the new Appendix A. To briefly summarize, we show that the neuron symmetry group (i.e. the symmetry in the parameter space) is a subgroup of the symmetric group (i.e. the symmetry for the neural graph). This implies that any model that is equivariant with respect to the symmetric group is automatically equivariant with respect to the symmetries in the parameter space. Hence, GNNs and Transformers operating on graphs are equivariant.
>
> > Why do the MNIST dilation numerical results differ so much from those in the Zhou et al. paper? They achieve about .070, whereas you achieve about .02.
>
> The results differ because we use the MNIST INR dataset that is created and publicly shared by Navon et al. [1] (i.e. the same one that we used for the classification setting). In that dataset, the INRs are trained to predict an intensity value in the range of [0, 1], whereas the dataset created by Zhou et al. [2] trains the INRs with a target range of [-1, 1]. This difference in the target range implies a different scale for the mean-squared error.
>
> For completeness, we also retrain our GNN and Transformer using the MNIST variant from Zhou et al. NFN has an MSE of 0.068, while our NG-GNN has 0.0635$\pm$0.0002 and our NG-T 0.0556$\pm$0.0004, consistently outperforming the NFN baseline.
>
> > Could you report how many probe features are used in each of your experiments?
>
> The INR experiments in Figure 3 show the number of probe features on the x-axis, ranging from 0 to 64. We also report the number of probe features in Appendix F. In the task of predicting CNN generalization and the learning to optimize task, we use 0 probe features. We clarify this in the updated text.
>
> ### References
> [1] Aviv Navon, Aviv Shamsian, Idan Achituve, Ethan Fetaya, Gal Chechik, and Haggai Maron. Equivariant Architectures for Learning in Deep Weight Spaces. ICML 2023.
>
> [2] Allan Zhou, Kaien Yang, Kaylee Burns, Yiding Jiang, Samuel Sokota, J Zico Kolter, and Chelsea Finn. Permutation Equivariant Neural Functionals. NeurIPS 2023.

---

### Author Response · Authors · 2023-11-23
**Author response to all reviewers**

We thank the reviewers for their thoughtful and constructive review of our manuscript. We were encouraged to hear the reviewers appreciate the novelty and flexibility of our method, as well as the thoroughness of our empirical evaluation.

Below we provide our detailed responses to each reviewer. We have also uploaded a revised version of our submission, with major changes highlighted with a different color in subsections 4.1, 4.2, and appendices A, B, C, and F.

---

### Meta-Review · Area_Chair_9eEj · 2023-12-05

**Metareview:**

In this paper, a very interesting approach is proposed, using graph machine learning to infer representations of other neural networks, while preserving equivariance properties. I find this a very interesting and timely contribution to an exciting body of "hypernetwork" style works, and GNNs are a natural and elegant candidate for this purpose. All Reviewers are in agreement, with several championing the work. I recommend acceptance without hesitation!

**Justification For Why Not Higher Score:**

N/A

**Justification For Why Not Lower Score:**

The paper offers a groundbreaking yet elegant approach to a very important problem (learning representations of neural networks). Two reviewers champion the work for an oral, which I have never witnessed in my previous service as an AC. I fully concur with the reviewers.

---

### Decision · Program_Chairs · 2024-01-16

Accept (oral)